# Dynamical similarity analysis can identify compositional dynamics developing in RNNs

## Abstract

Methods for analyzing representations in neural systems have become a popular tool in both neuroscience and mechanistic interpretability. Having measures to compare how similar activations of neurons are across conditions, architectures, and species, gives us a scalable way of learning how information is transformed within different neural networks. In contrast to this trend, recent investigations have revealed how some metrics can respond to spurious signals and hence give misleading results. To identify the most reliable metric and understand how measures could be improved, it is going to be important to identify specific test cases which can serve as benchmarks. Here we propose that the phenomena of compositional learning in recurrent neural networks (RNNs) allows us to build a test case for dynamical representation alignment metrics. By implementing this case, we show it enables us to test whether metrics can identify representations which gradually develop throughout learning and probe whether representations identified by metrics are relevant to computations executed by networks. By building both an attractor- and RNN-based test case, we show that the new Dynamical Similarity Analysis (DSA) is more noise robust and identifies behaviorally relevant representations more reliably than prior metrics (Procrustes, CKA). We also show how test cases can be used beyond evaluating metrics to study new architectures. Specifically, results from applying DSA to modern (Mamba) state space models, suggest that, in contrast to RNNs, these models may not exhibit changes to their recurrent dynamics due to their expressiveness. Overall, by developing test cases, we show DSA's exceptional ability to detect compositional dynamical motifs, thereby enhancing our understanding of how computations unfold in RNNs.

## 1 Introduction

Both neuroscience and mechanistic interpretability aim to understand how neural networks solve the problems they face (He et al., 2024; Lindsay & Bau, 2023; Vilas et al., 2024). As the architectural complexity of such models increases, there is a growing need to have tools which allow us to compare models across a wide array of task conditions and training parameters, at scale. One approach which has become popular is the use of representational alignment metrics which allow to conduct comparative analyses on the activations observed across networks (Sucholutsky et al., 2023) (Fig. 1). In mechanistic interpretability they can be used to compare how information is represented across layers (Raghu et al., 2021) and in neuroscience they allow us to test under which conditions neural networks work like specific brain regions (e.g. ventral stream of the brain; Kietzmann et al. (2019)).

While much of the literature is focused on static representations, meaning the activations observed in networks to non-dynamic stimuli like images, there has been an increasing interest in capturing the similarity of the dynamics of representations, so how representations change over the time course of a problem while the network is computing the correct response (Ju & Bassett, 2020) (Fig. 1). This is important due to the dynamic nature of both neuroscientific data (Vyas et al., 2020) and recurrently computing network models (Gu & Dao, 2023). Recent metrics like Dynamic Similarity Analysis (DSA) (Ostrow et al., 2024) address this challenge.

While the development of new metrics is exciting, new results have also shown that representational alignment metrics can respond to representations that are not computationally relevant to the system (Dujmović et al., 2023). At the same time, the choice of metric can impact results (Soni et al.,

2024). This suggests that researchers need to start developing sets of well-understood test cases which could serve as benchmarks to compare which metrics best capture relevant features in neural signals. In this way researchers could gradually refine metrics to respond to established test cases and hence become increasingly sure metrics capture the features in representations that are relevant for understanding neural computations (Klabunde et al., 2024; Ahlert et al., 2024).

In this work, we propose that we can use well-understood cases of simulated attractors and learning in recurrent neural networks (RNNs) to build test cases that assess how well metrics respond to dynamical representations while also testing whether such representations can be connected to the computations performed by networks. **Specifically, we construct two test cases: The first,** based on simulated attractor dynamics, allows us to assess in a controlled manner how robust similarity measures are in identifying dynamical motifs commonly observed during computations of stateful neural networks, in the presence of noise and when dynamical motifs are compositionally combined. This controlled setup allows us to evaluate metrics based on quantitative predictions. **The second test case**, builds on findings showing that RNNs learn to solve a compound task, made up of multiple subtasks, by compositionally combining the representational dynamics of subtasks (Driscoll et al., 2024). This test case allows us to make ordinal predictions for how

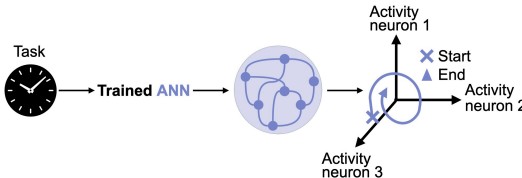

Figure 1: **Schematic showing the idea of dynamic representations**. A neural network (middle) is trained to solve a task which has a time course (left). We can now try to understand the internal mechanisms of that network by analyzing the representations of that network during task solving. For this, we record the activations of the network over time. These can be visualized by plotting the activation of all neurons as traces over time (right). These traces can now be compared across conditions. CKA and Procrustes only consider the geometric shape of the trace whereas new specialized metrics like DSA look at the traces as actual dynamics with momentum.

similar representations should become across compositional learning and creates the opportunity to link gradually developing representations with the resulting computations.

With these test cases, we can compare three metrics used to compare dynamic representations, namely Procrustes transformation (Procrustes), Centered Kernel Alignment (CKA), and Dynamic Similarity Analysis (DSA). **Our main contributions are showing that**:

- DSA shows better noise-robustness and ability to identify combined dynamics
- Out of the metrics we tested, DSA is the only measure which responds correctly to the compositional learning of the RNN-based test case
- DSA is the only measure that can link representations to computations by showing that increasingly similar representations are predictive of increasingly similar behavior in RNNs
- When applying DSA to state space models (SSMs) our results suggest that Mamba (Gu & Dao, 2023) might not require changes to dynamics during learning of simple tasks

## 2 RELATED WORK

A lot of progress has been made on using representational alignment metrics to both compare and induce similarities across models and empirical data (Sucholutsky et al., 2023; Williams et al., 2021). With representational metrics gaining popularity, researchers have identified the need to properly compare metrics to understand their shortcomings (Dujmović et al., 2023; Soni et al., 2024). Efforts to propose methods for benchmarking static representational alignment metrics aim to enable informed decisions about which metric is most suitable for a given application (Klabunde et al., 2024; Ahlert et al., 2024). Therefore, the goal is not necessarily to find "the best metric" but instead to highlight in which context to rely on a specific metric, similar to specialized statistical tests.

Our work specifically focuses on comparing dynamic representations (as opposed to static representations; see Fig. 1). While researchers have been interested in using representational alignment metrics for dynamic use cases (Lin et al., 2019; Williams et al., 2018; Kietzmann et al., 2019; Maheswaranathan et al., 2019; Cloos et al., 2024), they usually relied on extending static metrics to

also be applied to dynamic use cases. With growing interest in dynamic metrics, we are now seeing the development of new approaches to compare dynamical representations (Redman et al., 2024; Ostrow et al., 2024; Chen et al., 2024). Generally speaking, being specialized for dynamical representations means that metrics can actively capture the momentum of traces (i.e. the speed by which a system moves from point A in state space to point B in state space) in instead of just their shapes (i.e. only capturing whether the system moves from A to B at all). For example, DSA (Ostrow et al., 2024) compares the temporal dynamics of two systems by embedding them in a high-dimensional space and using a modified Procrustes analysis to assess the similarity of their vector fields. Another example is Diffeomorphic Vector Field Alignment (Chen et al., 2024) which evaluates the similarity by learning a nonlinear coordinate transformation that aligns vector fields, providing a measure of functional similarity through orbital equivalence. Although new metrics are being developed and applied (Eisen et al., 2024; Lipshutz et al., 2024), the challenge of determining when to use a specific metric remains an open question, especially in the context of dynamic metrics.

It has been argued that representations observed in neural systems should link to and hence help understand the computations executed by networks (Barack & Krakauer, 2021; Baker et al., 2022). This is linked to the idea of population codes in neural system and the Hopfieldian view of neural processing (Barack & Krakauer, 2021; Averbeck et al., 2006). Understanding how exactly representations and different neural codes link to computations has been an active focus of theoretical and empirical work (Tye et al., 2024; Fusi et al., 2016; Johnston et al., 2020; Dapello et al., 2022; Huber et al., 2023; Zheng et al., 2024). Visualizing the dynamics of representation in Euclidean space served researchers as an important tool to understand how the attractor dynamics observed recurrent systems link to their computations (Dubreuil et al., 2022; Vyas et al., 2020; Mante et al., 2013; Sussillo, 2014; Langdon et al., 2023). Following from this, dynamical representational alignment metrics ought to be a tool to identify and compare the attractor dynamics that link to the computations of recurrent systems (Baker et al., 2022).

In the following we address these two trends in the literature: we extend the idea of benchmarking representational alignment metrics to the dynamic case. By conceiving two test cases we can investigate whether representational metrics respond to the representations of a neural network that link to the network's computations. We do this by testing whether gradually evolving representations link to developing computational abilities throughout learning. These two isolated test cases could form part of a larger benchmark for dynamic representation alignment metrics in the future.

## 3 METHODS

**Metrics**

We use the three metrics: Centered Kernel Alignment ('CKA', Kornblith et al. (2019)), Procrustes Transformation ('Procrustes', Cloos et al. (2024)) and Dynamical Similarity Analysis ('DSA', Ostrow et al. (2024)). CKA and Procrustes were initially conceived for static representations and hence consider the dynamics as static traces and try to either align the two traces through optimal geometric transformation (Procrustes) or measure the similarity between the kernel matrices of the feature spaces (CKA). These metrics have shown promise in being adaptable to dynamic representations Cloos et al. (2024). In contrast, DSA is a metric specifically developed for dynamic representations and considers neuronal traces as dynamic, projecting them into high dimensional linear spaces where it compares the transition matrix of traces (Fig. 1). We use DSA with the hyperparameter *number of delays = 33* and *delay interval = 6*. Appendix 6.1 explains how parameters were set.

**RNN / SSM training and analysis**

For every condition, we trained 72 Recurrent Neural Networks (RNNs) and 64 State Space Models (SSMs) using the Mamba architecture (Gu & Dao, 2023) and the implementation by Torres-Leguet (2024). The hyperparameters for the RNNs and SSMs were chosen based on Driscoll et al. (2024). The training setup is expressed in more detail in Appendix 6.1.

**Tasks for RNN-based test case**

We used Neurogym task implementations to train RNNs (Molano-Mazon et al., 2022). Different task versions required networks to compare two stimuli, fixate during presentation and delay periods, and choose the higher or lower stimulus based on task-specific rules. Section 6.3 explains how task

versions are created. Appendix table 3 contains additional details. Network inputs included the standard task inputs alongside a one-hot task identifier for the current task (A, B, C, M).

# 4 EXPERIMENTS

Through the following experiments we want to test whether representational alignment metrics can capture the compositional representations and computations of stateful artificial neural networks. We specifically compare the performance of CKA, Procrustes, and DSA. The first test case is focused on simulated attractor dynamics that mirror dynamics usually observed to be underlying computations in RNNs. In the second case we analyze how identified dynamics with computations of trained RNNs. We close by applying metrics to analyze newly developed SSMs. The code used in this study will be made available on GitHub for reproducibility and further exploration. Appendix 6.1 provides additional information on the methods used for the metrics and the training schedules.

## 4.1 METRICS' ABILITY TO IDENTIFY COMPOSITIONALLY-COMBINED NOISY ATTRACTORS DYNAMICS

First, we want to test how well metrics capture attractor dynamics in the presence of noise and when multiple attractor dynamics are combined compositionally. We see these as core skills needed for metrics to perform in the more complex RNN-base test case introduced later (6.3). To test for these abilities, we simulate attractor dynamics using Lorenz attractors ($\sigma = 10$; $\beta = 2.667$).

All metrics can identify basic attractor motifs (see Appendix 6.2) but how well can metrics identify attractors in the presence of noise? For this, we construct two 'noise + attractor' combinations ('models') (Fig. 2a and Fig 8) where "+" refers to a numerical addition and scaling back to the unit volume. Model 1 combines one attractor sample (Attractor A) with one sample of Gaussian noise (Noise A). Model 2 combines the same attractor sample (Attractor A) with a different sample of Gaussian noise (Noise 2). We now gradually reduce the noise level across both noise samples in three steps ('Combinations'), by 50% each time, and compare Model 1 with Model 2 within each set of Combinations. We sample 7 sections of different Lorenz attractors (see Appendix 6.2), and each is used once as 'Attractor A'. Each attractor is used 200 times with different noise samples. Hence we run a total of 1400 comparisons. Results are depicted in Fig. 2b. As we want metrics to respond to the true underlying dynamic and ignore the noise, we want to see that across epochs, the similarity is a horizontal line, so not changing in response to changing noise levels. This is measured by a low dissimilarity gap (difference in dissimilarity between epoch 1 and 3). DSA shows the lowest response to noise ($1.7 \times 10^{-2}$ vs $4.5 \times 10^{-2}$ [CKA] vs $3.9 \times 10^{-2}$ [Procrustes]).

To identify compositional and computationally-relevant dynamics, metrics not only need to be noise robust but specifically noise robust in the context of compositionally-combined attractor motifs. To test for this ability, we assemble 'Model 3' which combines Attractor A with another Attractor B and a new sample of Gaussian noise (Noise 3). For Model 3, we gradually decrease both the noise and the relative amplitude of Attractor B. We use the same 7 attractor samples as above, so we get 42 attractor pairings where Attractor A and Attractor B are not the same. Each attractor pairing is used 200 times with different noise samples. Hence we run a total of 8400 comparisons. How do we know that a metric responds correctly to the combination of attractors? Given that we use random combinations of attractors and noise, with a linearly decreasing influence of the additional attractor and noise, we would like to see a linear decrease in dissimilarity across epochs as measured by the linearity measure. We assess the linear decrease of dissimilarity by computing the $R^2$ of the linear fit of a regression over the dissimilarity predicted by the epoch (Fig. 2c). DSA shows the most linear decrease (0.99 vs 0.96 [CKA] vs. 0.97 [Procrustes]). Note that one might want metrics to behave like a ratio to a test case like this. A measure behaves as a ratio if it has equal intervals between values, and has a meaningful zero point, allowing for meaningful ratios between any measurements. In our case that would mean metrics should decrease linearly and show a value of 0 for full similarity. Fig. 2b highlights that while DSA shows the least change to noise, it is furthest away from true zero. This results in DSA having a lower dissimilarity gap in Fig. 2c. Both Procrustes and CKA get closer to true zero. This highlights that DSA might need additional normalization to function like a ratio.

These analyses based on simple and well-controlled attractor dynamics reveal that DSA might have a very slight advantage at identifying dynamical motifs in the presence of noise and when they are

compositionally combined. From these analyses alone it is not possible to infer whether these very minor quantitative difference result in qualitatively different behavior of metrics in more complex environments. The next section will hence explicitly focus on a more complex test case.

## 4.2 Metrics' ability to identify compositional dynamics during over training in RNNs

Metrics for describing representations are naturally most useful for cases where the amount of data to be analyzed is too large to allow for fully manual description, because of long time courses and model dimensionality. As such, we next want to move to a more complex test case that still allows us to make specific predictions about the expected relationships of representations across models. To construct this test case, we build on analyses of RNNs trained to learn simple cognitive tasks commonly used in computational neuroscience (Barbosa et al., 2023; Yang et al., 2019; Molano-Mazón et al., 2023; Proca et al., 2024; Aitken & Mihalas, 2023) and implemented in the Neurogym library (Molano-Mazon et al., 2022). More specifically, we build on results showing how RNNs learn compound tasks in a compositional fashion (Driscoll et al., 2024; Yang et al., 2019).

In their work, Driscoll et al. (2024) show that when a task is constructed as a combination of subtasks, the developing overall representation similarly looks like a combination of the representations of these subtasks. As a specific example, the 'Delay task' (Fig. 3a) requires networks to observe two streams of continual inputs ('Stimulus Modality 1' and 'Stimulus Modality 2') while the fixation signal is on, and then choose the stimulus with the higher average value once the fixation value disappears. There is a delay between the end of the stimulus period and the end of the fixation signal, during which the RNN needs to remember the response. Driscoll et al. (2024) make use of the fact that these simple tasks can easily be combined. For example, the 'Delay task' can be combined with an 'Anti task' which results in a

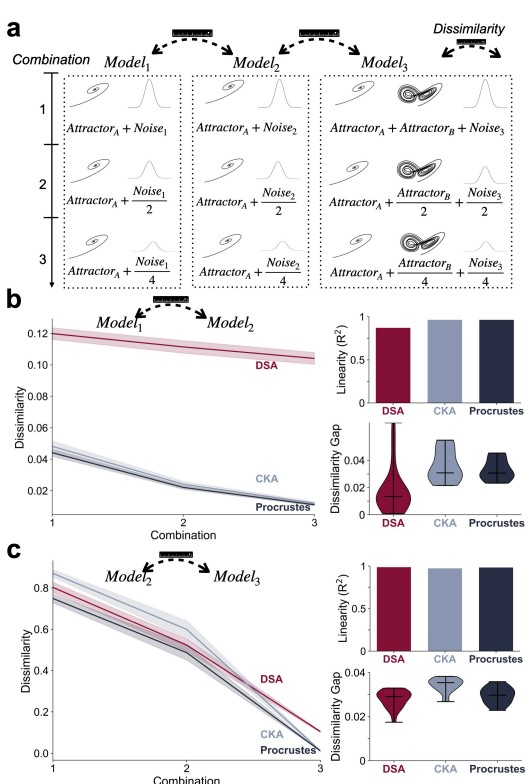

Figure 2: **DSA shows better noise robustness and identification of compositionally-combined dynamics.** (a) Outline of model specifications to test both noise robustness (Model 1 vs. Model 2) and identification of combined dynamics (Model 2 vs. Model 3). (b) Results of noise robustness (c) compositional dynamics comparisons. All shadings are standard errors.

'DelayAnti' task where network do the same as in the 'Delay task' but the choice at the end of a trial is towards the weaker stimulus. They observed that the representational dynamics within networks while solving a compound task is a combination of the dynamics of both separate tasks. So, the dynamics of 'Delay' (Fig. 3c) and 'Anti' (Fig. 3b) combine to 'DelayAnti' (Fig. 3d). To show this they used empirical bifurcation diagrams with fixed-point analyses. They also observed that if they created a compound task, the performance of networks when solving the compound task would increase with the number of subtasks they were pretrained on (Fig. 3e). Their results give us a very specific analytical understanding of how networks combine motifs to solve these tasks.

How can we use this knowledge to test representational alignment metrics? The work outlined above makes the specific prediction about how the similarity of RNNs' representations develops as an overlap of the training schedule. To test the metrics, we can replicate the training setup by Driscoll et al. (2024) and see whether metrics identify these similarities of representations. This is depicted in Fig. 3f. The baseline RNN is freely trained on a compound 'Master task' (called 'M' on figures) without any pretraining (top row Fig. 3f). This 'Master' network (red) can freely learn all dynamical

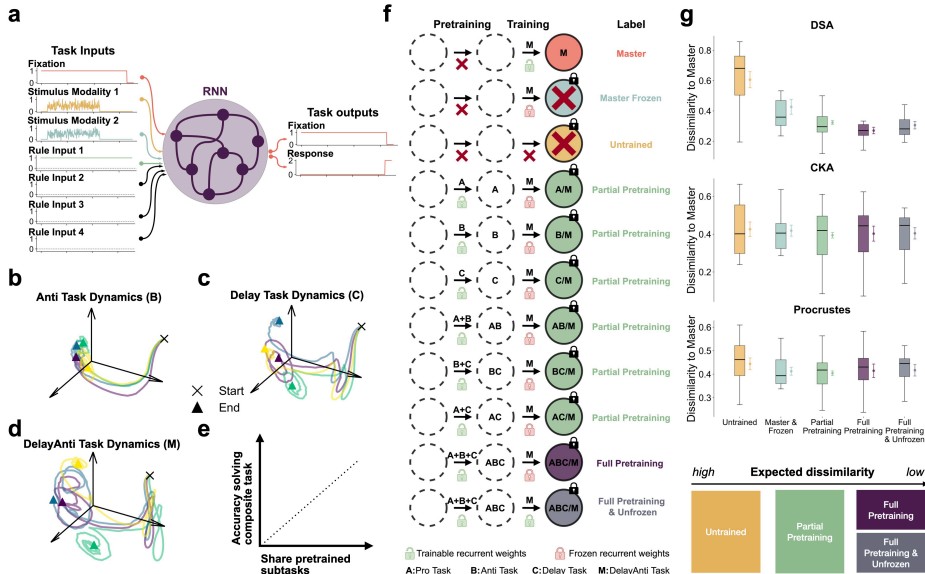

Figure 3: **DSA, but not CKA or Procrustes, captures how representations develop during compositional learning.** (a) RNN inputs and outputs in our setup. (b) Schematic of dynamical representations during 'Anti task'. Different colors show different task conditions. (c) Same as (b) but for the 'Delay task'. (d) Same as (c) but showing the compound (Master) 'DelayAnti task'. (e) Schematic of observation in (Driscoll et al., 2024). (f) List of training conditions used in our test case. (g) We calculate the dissimilarity of all training groups to the 'Master' group with results shown by training schedule and metric. Color of boxplots refers to color used for training conditions in (f). Order at the bottom of the plot highlights the expected dissimilarity. Lines next to boxplots are standard errors.

motifs it needs to solve the task with its recurrent weights. We then also have a set of networks which are pretrained on partial sets of the subtasks and later trained on the Master task, but all their weights except for input weights are frozen between pretraining and training ('Partial pretraining' networks; green). These networks can only partially learn the needed dynamics in their recurrent weights and need to use their input weights for otherwise missing recurrent transformations. As stated before, Driscoll and colleagues observed that these networks only partially developed the dynamics of the Master network. We build additional special cases which are fully pretrained on all subtasks and are either frozen or not frozen after pretraining (purple and grey respectively). We also add networks which are only allowed to use their input weights to learn the Master task, but without any pretraining ('Master frozen'; light blue). Lastly, we add a group of untrained networks which only observe the inputs with purely random weights ('Untrained'; yellow). For each group of networks listed in Fig. 3f we train each of the 72 RNN with different hyperparameters (see Methods 6.1).

What are the specific predictions for how a metric should behave? Based on the prior results we can make ordinal predictions about how similar the representations of each network group should be to the representations in the 'Master' group (Fig. 3g, bottom). The 'Master' group is here the reference group and represents the networks which can learn the compound task without constrains. As the computational structure of the tasks is known, we expect a higher alignment (with the 'Master' group) of the conditions trained with the highest share of subtasks. We would expect 'Untrained' to be the least similar, followed by 'Partial pretraining', followed by both fully pretrained groups (with no specific expected order between these two). Prior results do not allow for strong predictions about the 'Master & frozen' group. If metrics cannot identify this order then this would lead us to conclude that they cannot uncover the computational structure of the tasks and thus not be able to correctly detect the development of representations in the networks.

To test whether metrics match predictions, we measure the dissimilarity between the networks of each training group and their corresponding network in the 'Master' group. Note that prior results (Maheswaranathan et al., 2019) show that the network parameters such as activation functions can have strong influences on the dynamics, so that we compare similarities between networks with

the same hyperparameters. The results of these analyses are shown in Fig. 3g for all metrics. We observe that only DSA shows the order we expect to see based on prior results. Calculating pairwise significance tests between all groups with the dissimilarity distribution of 'Full Pretraining' shows that all groups are significantly different from this baseline, except for 'Full Pretraining & Unfrozen' (all p-values FDR corrected; values in Appendix 6.3 Table 5). These results suggest that DSA and CKA do not identify meaningful differences between groups. This statement stays true even when comparing all the groups against each other and not only to 'Master' (Appendix Fig 9).

The analyses of this section reveal that DSA is the only metric which correctly identifies the compositional representation we expect to see in RNNs. At the same time, we show that the test case developed here is a nontrivial challenge for representational metrics, as both CKA and Procrustes fail to correctly discriminate between training schedules.

### 4.3 TESTING METRICS TO IDENTIFY TASK RELATED COMPUTATIONS ALONGSIDE TASK RELATED DYNAMICS

While our first analysis of the RNN-based test case presented a non-trivial challenge to metrics, it falls short of linking representations to computations

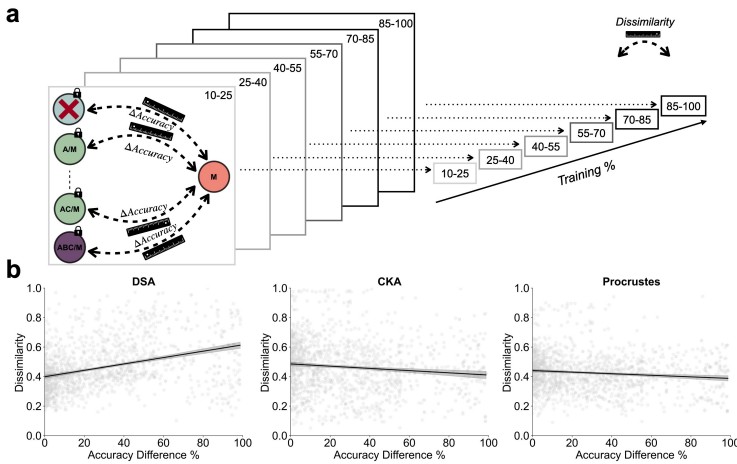

Figure 4: **Only DSA can link developing representations to developing computations.** (a) Schematic of analysis. As before, we calculate the dissimilarity between all training groups and the 'Master' group, but we now also capture the difference in accuracy between the two networks used for the dissimilarity calculation. We measure the dissimilarity and accuracy at multiple windows during training. (b) Results from analysis in (a) plotted by dissimilarity measure. All shadings are standard errors.

which is discussed a key criteria of representations (Baker et al., 2022). Representations link to computations if they relate to the full transformation between the inputs and the corresponding outputs (the choice made by the network). In contrast, metrics could capture dynamics which transform network inputs in a 'null-space' (Kaufman et al., 2014) that ultimately does not translate into the choice space. This can lead to networks with different computations showing seemingly similar representations (Dujmović et al., 2023).

We now want to use our RNN test case to see whether metrics directly link to the computational process of the network. The simplest way to summarize the differential computations of two networks is to calculate their difference in task accuracy. Networks with more similar task accuracy likely implement more similar computations than networks with less similar task accuracy, especially in simple tasks such as ours where there are not multiple different strategies which allow for a correct solution. Hence, for a metric to capture networks' computations, we expect a higher similarity in representations of two networks to be predictive of a lower gap in accuracy. To test this, we measure the dissimilarity between the 'Master' group and any other group as before. For each comparison of networks, we also capture their difference in task accuracy on a validation set of trials. Instead of just doing this at the end of training, we are now capturing the networks during six windows of training (10-25%, 25-40%, ..., 85-100% of training based on epoch number; Fig. 4a) and comparisons of dissimilarity are done within these blocks of training. Within a group of training progression, we take the median measure of accuracy and dissimilarity as the measure of that group. This results in 3456 pairs of dissimilarity and accuracy values (72 network parameter setups, 8 training schedules, 6 comparisons across time, all multiplied) for each metric. We expect that a higher level of dissimilarity will be linked to a higher disparity in accuracy. Fig. 4b shows these relationships split by metric. We observe that, again, only DSA shows the expected pattern, with a significant positive relationship between these variables ($b = 0.22, p = 3.3 \times 10^{-47}, R^2 = 0.12$ ; other regressions in

Appendix 6.4 Table 6). We do observe that even networks with the same accuracy value (i.e. 0% accuracy difference) can still differ in their representations, which is likely caused by the difference in their training schedule. These analyses further strengthen the case for DSA. Not only does it behave as predicted to training schedules, but it also shows evidence of linking directly to the computations that networks are executing. Our test case here shows a capability of DSA that is difficult to highlight in test cases which only use simulated and non-task-solving dynamics.

### 4.4 METRICS' RESPONSES TO INCREASING OVERLAP OF THE TRAINING SCHEDULE, ACROSS THE DURATION OF LEARNING

With the new RNN test case we were able to show that DSA shows expected ordinal responses to training schedules and can link representational dynamics to computations. This tested expectations based on Driscoll et al. (2024). Using representational metrics carries the promise that one can precisely quantify the relationship between dynamics, instead of just relying on qualitative observation. In this section we want to demonstrate this by testing whether an increasing overlap in training schedule also causes an increasing alignment of representations. Additionally, we want to observe how this alignment develops over training. The following analysis are exploratory and that analyses by Driscoll et al. (2024) do not make specific predictions for these cases.

We start with testing for the effect of a gradually overlapping training schedule on representations. To test this, we run a full set of pairwise comparisons between networks, meaning each network from one specific training schedule is compared with networks of every other training schedule (Fig. 5a). For each network comparison, we quantify to which degree the two networks overlap in their training schedule. For example, a network pretrained on task A before being trained on compound task M, would share 50% with a network trained only on M and 66% with a network trained on A, C, and M. Fig. 5a highlight example comparisons. As before, we only ever compare networks with the same hyperparameters. For this analysis we only use networks that have completed training.

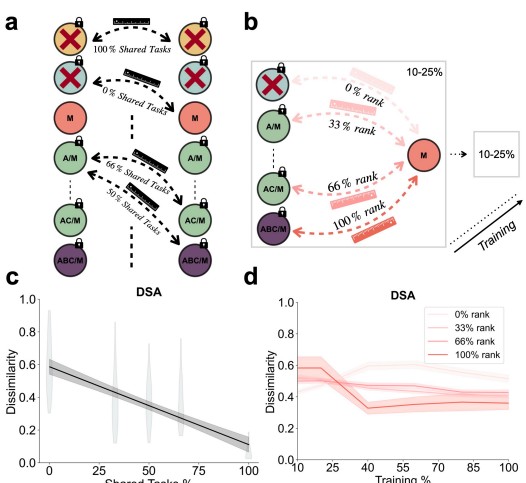

Figure 5: **DSA responds to gradual increase in task overlap during training.** (a) Schematic showing how every training group is compared to every other training group to capture how the % of shared tasks during training affects dissimilarity. (b) Schematic showing how the analysis from prior section (Fig 4a) is adapted to capture the share of pretraining tasks that a network has 'experienced' effects dissimilarity (called 'rank' to differentiate from terminology in Fig 5a). Schematic only shows 10-25% group for visual simplicity, but all six time groups from Fig 4a are analyzed. (c) DSA results of analysis in Fig 5a. (d) DSA results of analysis in Fig 5b. All shadings are standard errors.

The results depicted in Fig. 5c show that DSA can recognize the difference between full overlap of tasks, partial overlap of tasks, and no overlap of tasks, but does not seem to identify a gradual increase in overlap within the groups which have a partial overlap in tasks. CKA and Procrustes behave similarly (Appendix Fig. 9a and Fig. 9b). All regression parameters are in Appendix 6.5 Table 7.

We additionally want to assess how representations develop over the course of learning. For this we use the prior analysis of comparing every training setup to the 'Master' setup over training time (Fig. 5b). We then plot dissimilarity values over training time, grouped by the pretraining 'rank', meaning how many of the subtasks making up the Master task were included in a network's pretraining. Being not pretrained would be 0%, whereas being pretrained on B and C would make 66% (Fig. 5b). In this analysis we use all possible pretrained model configurations and so we use the blue, green and purple groups from Fig 3f . Fig. 5d shows the results for DSA. The 100% line shows that the reference model and the Master model become more similar as the Master learns the task, and hence is closer to the computations conducted by the pretrained model than the 'Partial

Pretraining' group. This pattern is inverted for networks that have 0% task overlap (green): they start out similar when neither have learned to solve the task but become less similar as the master network learns. We again do not observe a gradual difference between different groups of partial overlapping training schedules. Other metrics are not reacting to this signal (Appendix Fig. 9).

### 4.5 USING DSA AND THE ESTABLISHED TEST CASE TO ANALYZE THE LEARNING PROCESS OF STATE SPACE MODELS

Above we showed how test cases based on well-understood empirical phenomena can be used to compare the ability of different metrics to extract computationally relevant representations across learning. Next, we want to show that such a test case can also be used in the reverse to study how architectural changes effect learning. For the case of stateful networks we have seen new architectures being released during the last year and so we want to test whether the newly introduced Mamba architecture (Gu & Dao, 2023) learns tasks in the same way that RNNs do.

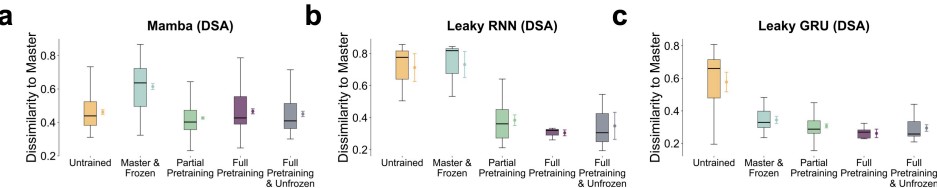

Figure 6: **DSA suggests more reservoir-like learning in Mamba when compared to RNNs.** (a) Results of running the analysis from Fig. 3g with the Mamba architecture and DSA. (b) DSA results from Fig. 3g but now only showing Leaky RNNs. (c) DSA results from Fig. 3g but now only showing Leaky GRU. Lines next to boxplots are standard errors.

To compare whether SSMs learn like RNNs, we apply the same test case we applied to RNNs in Fig. 3 to SSMs. As with RNNs, we use a wide set of network parameters (see Methods) and compare across training schedules (Fig. 3f) but within hyperparameters. We observe that Mamba learns quicker (2.7 times quicker than RNNs on the Master task) and models converge more reliably (99% convergence vs. 91% convergence on Master task). The results of the representational dissimilarity analyses are depicted in Fig. 6a, showing the same analysis as in RNNs (same plot as Fig. 3g). Mamba shows that the group that was trained on the Master task but with all weights frozen, except for the input weights, is the most different from representations observed in the Master network ($p = 2.7 \times 10^{-16}$; other p values in Appendix 6.6 Table 10). All other groups seem to roughly be in the same range of similarity to Master, though note 'Partial Pretraining' is also significantly more like the Master network then the 'Full Pretraining' is ($p = 6.2 \times 10^{-4}$). Regardless, 'Master & Frozen' clearly sticks out as the most different group. This means that all training groups produce roughly the same dynamics except for the group which is only allowed to learn the task through its input weights. This suggests that the very expressive Mamba architecture, when trained without freezing on a simple task such as ours, does not learn by changing its hidden state dynamics but instead mostly learns through optimizing the transformation of the out-read layer (i.e. the last MLP layer). Given that the hidden state dynamics of most trained network look like the dynamics of the untrained network (p-values given in Appendix 6.6 Table 11) would lead to the prediction that the hidden state of Mamba during this simple task functions like the reservoir of a reservoir network (Lukoševičius & Jaeger, 2009). As before, this is a pattern only identified by DSA (see Appendix 6.6 Fig. 10 for CKA and Procrustes). Additional control analyses show that the RNN pattern from Fig. 3g holds true even when splitting the results into the RNN subtypes (Fig. 6b and Fig. 6c).

## 5 DISCUSSION

Representational alignment metrics have become popular for comparing representations in neural systems across architectures and conditions. While they can be a helpful and scalable tool, recent work also has identified potential pitfalls when they react to spurious signals (Dujmović et al., 2023; Soni et al., 2024). Here we introduce two test cases which can serve as a first step towards a benchmark for dynamical representation alignment metrics. Using these to compare DSA, CKA, and

Procrustes reveals that DSA is the only metric that can identify how compositional representations emerge throughout learning and how these are then linked to task behaviours. Building on the combination of our test case and DSA also allows us to form a specific hypothesis about reservoir-like learning in Mamba models confronted with simple tasks.

We put forward two specific test cases, where the simpler attractor-based test case is aimed at fundamental skills needed for the more complex test case. These cases naturally do not fully validate DSA and hence we shy away from calling our work a full benchmark. Instead, we see this as a first step into the direction of identifying a set of well-established empirical finding which can then form a full benchmark for dynamical representation metrics. In this process, the researchers could carefully choose what they want (or do not want) metrics to capture. We believe this step of gradually collecting test cases will be necessary to generate a better understanding of what metrics can and cannot do. There likely is going to be a trade-off between the complexity of test cases and the precision of predictions that can be made about representations, similar to what we observe in our two cases. We hope that our example can spark a productive back and forth between finding test cases and improving measures that will ultimately increase researchers' confidence in the metrics. As the set of test cases increases, we will also increasingly be sure that metrics will generalize to new and unseen cases. Until then, the risk of overfitting to our specific cases seems low, as the quantitative predictions of the first case are already captured by all metrics and the qualitative predictions of the second case intrinsically do not allow for overfitting, as there is no specific metric to overfit to.

We also show that combinations of test cases with (partially) validated metrics can become important tools for studying architectural changes. Using our RNN-based test case, we can generate two new observations: In RNNs we seem to find that the development of representations in networks as a function of partial pretraining does not seem to be gradual but instead blocked into relatively discrete groups of not-pretrained, partially pretrained, and fully-trained. Additionally, through applying our case to Mamba we observe results which suggest that Mamba learns the simple tasks of our test case in a reservoir-like way, meaning that it does not strongly change its very expressive hidden state dynamics throughout learning and instead seems to use the layers following the hidden state for task related learning. This is unless we restrain the models so that they can only learn with their input weights, which then has a measurable effect in terms of changes to hidden state dynamics. Due to the exploratory character of these analyses, our results still need to be confirmed with more detailed analysis. If confirmed with further analysis, this would gradually increase our trust in these metrics.

## 5.1 LIMITATIONS

While here we focus on DSA, Diffeomorphic vector field alignment (Chen et al., 2024) has been introduced since. We could test whether it also holds up to this test case like DSA. With regards to the RNN-based test case we use a broad stroke measure of 'accuracy' to capture the networks' behavior. While that is enough to differentiate between DSA and Procrustes / CKA, a more nuanced view on behavioral strategies might be needed to compare metrics in the future. We also do not use any empirical data in this work, even though comparison to data is a use case of these metrics within neuroscience. Lastly, our work does not specifically identify why Procrustes / CKA perform worse than DSA. The attractor-analyses show better noise robustness of DSA, which in turn might help DSA to identify the true dynamics in the RNN-based test case. While plausible, these are speculations and we do not give a detailed analysis of why DSA outperforms other metrics.

## 5.2 CONCLUSIONS

Our tests identify DSA, out of the metrics we considered, as the most powerful dynamic representation alignment metric, providing specific evidence pointing towards its use when studying dynamical representations. Constructing additional test cases for representational metrics will likely help to gradually improve our understanding of these metrics and can help us to generate new hypotheses about how different network architectures learn and function.

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
