# 6 APPENDIX

## 6.1 METHODS

**Setting DSA hyperparameters**

Unlike Procrustes and CKA, DSA has hyperparameters. We use the computationally cheaper attractor analysis to sample a wide space of parameters and decide on the best parameter set. We then use the same parameters for the analyses of RNNs and SSMs. For the attractor analysis the "number of delays" parameter is sampled uniformly from 1 to 100. The "delay intervals" parameter is sampled uniformly from 1 to the number of time steps (fixed to 200) divided by the selected number of delays. We manually conduct an iterative search to find the best parameters. For each parameter, we take 10 samples within each interval. We refine the interval for each parameter two times, taking the two best candidates for each parameter as bounds for the new interval. We then select the best parameter combination based on best dissimilarity gap and linear behavior in analysis shown in Fig. 2e (number of delays = 33, delay interval = 6). Generally, the dissimilarity gap and linear behavior were correlated, so that a parameter combination doing well one measure, also did well on the other. As a result we did not have to trade-off between these two criteria. Note that the attractor analysis we run purposefully uses the same number of time steps as the analysis window in later RNN analysis. Generally speaking, we found that changing the DSA parameters does quantitatively but not qualitatively change results of analyses.

**RNN / SSM training and analysis**

The training process is detailed below:

| Parameter | Description |
|---|---|
| Architecture | Leaky RNN, Leaky GRU |
| Activation Function | ReLU, Softplus, Tanh |
| Hidden Size | 128, 256 |
| Learning Rate | $1 \times 10^{-2}, 1 \times 10^{-3}$ |
| Batch Size | 64, 128, 256 |
| Optimizer | Adam |
| Loss Function | Cross Entropy |
| Training Data | One epoch consisted of 10,000 trials for each task. |
| Training Procedure | Networks were trained in a supervised fashion until they reached 99% accuracy on each task or for 50 epochs, whichever came first. When a network was pretrained on multiple tasks, the tasks were trained sequentially within each epoch. |
| Analysis | We only analyzed the networks that managed to fully learn the master task to 99% accuracy, which corresponded to 91% of all models. |
| Training Time | The total training time for all conditions was 48 hours. |

Table 1: RNN Training Parameters

| Parameter | Description |
|---|---|
| Architecture | Mamba SSMs (Gu & Dao, 2023) |
| Number of Layers | 1, 2 |
| Hidden Dimensions | 8, 16 |
| Learning Rate | $1 \times 10^{-2}, 5 \times 10^{-3}, 1 \times 10^{-3}, 5 \times 10^{-4}$ |
| Batch Size | 16, 32, 64, 128 |
| Optimizer | Adam |
| Loss Function | Cross Entropy |
| Training Data | One epoch consisted of 10,000 trials for each task. |
| Training Procedure | Networks were trained in a supervised fashion until they reached 99% accuracy on each task or for 50 epochs, whichever came first. When a network was pretrained on multiple tasks, the tasks were trained sequentially within each epoch. |
| Analysis | We only analyzed the networks that managed to fully learn the master task, which corresponded to 99% of all models. |
| Training Time | The total training time for all conditions was 20 hours. |

Table 2: Training Parameters for Mamba SSMs

**RNN tasks**

Each task in Table 3 consisted of a stimulus presentation period (200 time steps) and a choice period (25 time steps), with an optional delay period as described above. The duration of the delay was variable during training (25, 50, or 75 time steps) but fixed during testing (100 time steps). We analyzed the hidden states during the stimulus presentation period, keeping the first twenty principal components after centering and normalization. Networks were optimized for correct fixation during stimulus presentation and delay periods, as well as correct choices after the fixation. Accuracy was reported based on the choice made by the network during the response period at the end of the trial, weighted towards the last time step.

| Task | Description |
|---|---|
| Task A (Pro Task) | The model received two continuous numbers as separate inputs with time-varying Gaussian noise. It had to decide which input was higher on average. During stimulus presentation, the inputs were encoded using a noisy sinusoidal representation to ensure non-trivial feature extraction. |
| Task B (Anti Task) | Similar to Task A, but the model had to decide which input was lower. The inversion required the model to learn an orthogonal decision process compared to Task A, emphasizing different representational strategies. |
| Task C (Delay Task) | Similar to Task A, but with an additional delay period before the decision phase. The delay period introduced a memory component that required the model to retain stimulus information for a variable amount of time before responding. |
| Master Task (M, DelayAnti Task) | A compound task in which networks had to determine which stimulus was lower after a delay period. This task combined elements of memory retention, delayed decision making, and anti-response logic, testing the network's ability to generalize and adapt across combined task features. |

Table 3: Tasks description

**Evaluation Metrics**

The performance of the models was evaluated using weighted accuracy, which took into account the mask applied to the predictions. The weighted accuracy was calculated by first applying a mask. The mask was used to weigh the correct predictions. The mask values progressively increase from 1

to 5 during the response period and were 1 otherwise. Finally, the weighted accuracy was calculated as the sum of weighted correct predictions divided by the total weight. This ensured that only the relevant predictions contributed to the accuracy metric.

**Hardware and Software Environment**

The training and analysis were conducted in a cluster computing environment using 8 NVIDIA Tesla V100 GPUs. The software environment is described in Table 4.

| Software | Version |
|---|---|
| Python | 3.8 |
| PyTorch | 1.9.0 |
| NumPy | 1.21.2 |
| SciPy | 1.7.1 |
| Matplotlib | 3.4.3 |
| Jupyter | 1.0.0 |
| TensorBoard | 2.6.0 |

Table 4: Software Dependencies

This environment ensured reproducibility and consistency across different runs and experiments.

### 6.2 ATTRACTORS

In an additional analysis we tested how well different metrics could differentiate the different kind of Lorenz attractors (i.e. one stable fix point, two stable fix points, two unstable fix points). We sample 9 examples of attractor dynamics each with 200 time-steps and 200 trials and pairwise compare each sampled attractor with every other attractor using all three metrics, generating 81 dissimilarity values per metric (Figure 7). We then summarize these by summing all values belonging to comparisons within a group of attractors (i.e. comparison with "one stable fix point" to another "one stable fix point") and summing all values belonging to comparisons across groups of attractors (i.e. comparison with "two unstable fix points" to another "one stable fix point"). The results of this are depicted in Figure 7b. We see that all measures can recognize whether two attractors belong to the same group or not, but DSA seems to perform slightly better in discriminating cases (average dissimilarity gap between within and across groups of 0.26 for DSA compared to 0.11 for CKA and 0.13 for Procrustes as well as non-overlapping box plots for within and across for DSA).

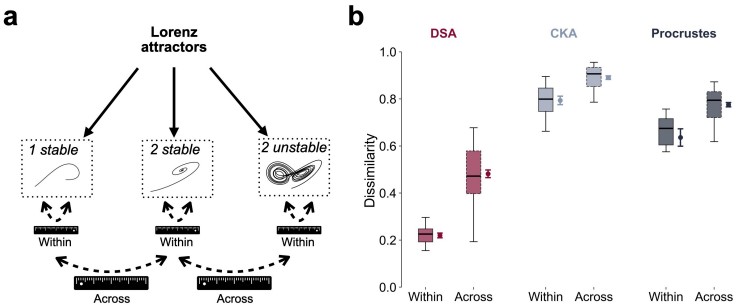

Figure 7: **All metrics can identify basic attractor motifs.** (a): Outline of attractor specifications to test the identification of motifs belonging to similar (Within) or different (Across). (b) Results of the identification.

This analysis allows us to choose the different Lorenz attractors we use for the analysis of Fig. 2a to test the 'ratio-response' of metrics to noisy combined attractors. To combine the Attractor A and B, we need them to be sufficiently dissimilar so that a different way of learning is simulated with Model 3. We thus choose 1 attractor in the '1 stable' group of Fig 7a. and 3 from each of the two other groups, as the '1 stable' group representatives are too similar to each other (all look like a line). The noise for attractors was centered Gaussian noise, with the standard deviation decreasing from 0.01 to 0.0025 over time, as the dynamics were contained within the unit sphere.

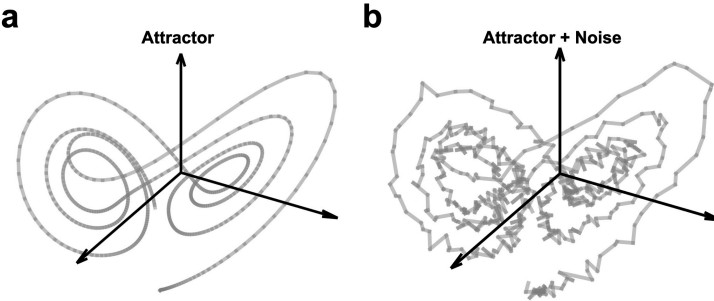

Figure 8: **Sample attractor with gaussian noise** (a): Sample attractor from the '2 unstable' group from Fig 7. (b) Same attractor with 1% of standard normal noise (corresponding to $Noise_1 = \mathcal{N}(0, 0.01^2)$ from Fig 2 for instance.

## 6.3 RNN SIMILARITY ANALYSIS WITH WITH MULTIPLE REFERENCE GROUPS

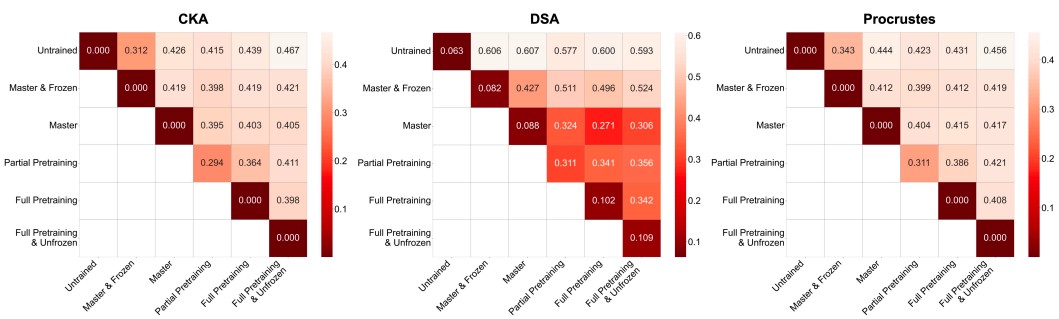

Figure 9: **CKA and Procrustes analyses do not distinguish between pretrained and untrained networks**. Median dissimilarity of all training groups against each other for all metrics. A darker red refers to a lower dissimilarity.

While Fig 3g showed the dissimilarity of all groups against the 'Master' group, we extended the analysis to all groups against each other in Fig 9. Here again, we see that DSA is the only metric which correctly identifies the expected compositional representation in RNNs. Besides the comparison to the 'Master' group, Fig 9 shows for instance that the pattern stays the same if we take the 'Full pretraining' group as a reference group. The dissimilarity is still low against 'Master' (0.271). However, it increases when computed against groups with an incomplete pretraining ('Partial Pretraining', 0.341) and further increases when compared against groups with no training at all ('Untrained', 0.600). However, CKA and Procrustes failed to correctly discriminate between different training schedules.

Table 5: Comparing the distributions of dissimilarity of each training group against the distributions of dissimilarity of the 'Full Pretraining' group (based on Fig. 3g). p-values from T-test corrected for multiple comparison with fdr-bh.

| METRIC | UNTRAINED | MASTER & FROZEN | PARTIAL PRETRAINING | FULL PRETRAINING & UNFROZEN |
|---|---|---|---|---|
| DSA | $2.5 \times 10^{-10}$ | $3.2 \times 10^{-4}$ | $1.6 \times 10^{-2}$ | $1.5 \times 10^{-1}$ |
| CKA | $9.2 \times 10^{-1}$ | $9.2 \times 10^{-1}$ | $9.2 \times 10^{-1}$ | $9.6 \times 10^{-1}$ |
| Procrustes | $7.1 \times 10^{-1}$ | $9.3 \times 10^{-1}$ | $8.7 \times 10^{-1}$ | $9.3 \times 10^{-1}$ |

## 6.4 REGRESSION TO IDENTIFY TASK RELATED COMPUTATIONS ALONGSIDE TASK RELATED DYNAMICS

Table 6: Parameters for regression predicting dissimilarity based on difference in accuracy (Fig. 4a).

| METRIC | Slope | Intercept | p-value of slope | $R^2$ |
|---|---|---|---|---|
| DSA | 0.22 | 0.40 | $3.3 \times 10^{-47}$ | 0.13 |
| CKA | $-0.08$ | 0.49 | $1.2 \times 10^{-5}$ | $1.3 \times 10^{-2}$ |
| Procrustes | $-0.05$ | 0.44 | $1.8 \times 10^{-5}$ | $1.2 \times 10^{-2}$ |

## 6.5 METRICS' RESPONSES TO INCREASING OVERLAP IN TRAINING SCHEDULE, ACROSS DURATION OF LEARNING

Table 7: Parameters of regression predicting dissimilarity based on % shared tasks (linked to Fig. 5c and Fig. 9a & b.

| METRIC | Slope | Intercept | p-value of slope | $R^2$ |
|---|---|---|---|---|
| DSA | $-4.8 \times 10^{-3}$ | $5.9 \times 10^{-1}$ | $1.1 \times 10^{-23}$ | 0.52 |
| CKA | $-3.5 \times 10^{-3}$ | $4.8 \times 10^{-1}$ | $2.1 \times 10^{-12}$ | 0.30 |
| Procrustes | $-3.5 \times 10^{-3}$ | $4.9 \times 10^{-1}$ | $6.2 \times 10^{-17}$ | 0.40 |

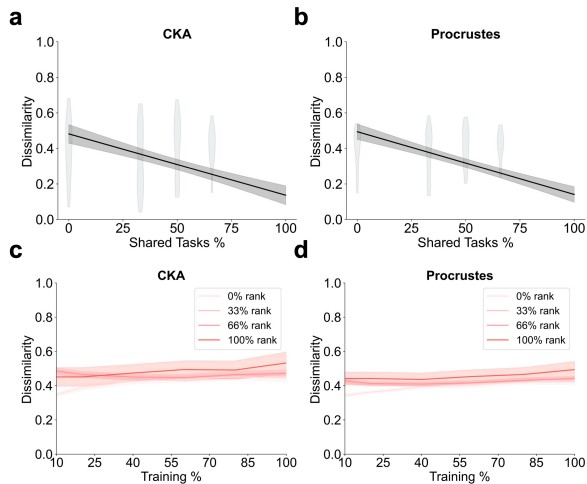

Figure 10: **CKA and Procrustes partially responds to gradual increase in task overlap during training.** (a) CKA results of analysis in Fig. 5a. (b) Procrustes results of analysis in Fig. 5a. (c) CKA results of analysis in Fig 5b. (d) Procrustes results of analysis in Fig 5b.

## 6.6 SIGNIFICANCE OF ORDER OF TRAINING SETUPS FOR STATE SPACE MODELS

Table 8: LeakyGRU: Comparing the distributions of dissimilarity of each training group against the distributions of dissimilarity of the 'Full Pretraining' group (based on Fig. 6c). p-values from T-test corrected for multiple comparison with fdr-bh.

| METRIC | UNTRAINED | MASTER & FROZEN | PARTIAL PRETRAINING | FULL PRETRAINING & UNFROZEN |
|---|---|---|---|---|
| DSA | $2.2 \times 10^{-7}$ | $3.4 \times 10^{-3}$ | $4.6 \times 10^{-2}$ | $1.9 \times 10^{-1}$ |
| CKA | $9.6 \times 10^{-1}$ | $9.6 \times 10^{-1}$ | $9.6 \times 10^{-1}$ | $9.6 \times 10^{-1}$ |
| Procrustes | $8.9 \times 10^{-1}$ | $8.9 \times 10^{-1}$ | $8.9 \times 10^{-1}$ | $8.9 \times 10^{-1}$ |

Table 9: LeakyRNN: Comparing the distributions of dissimilarity of each training group against the distributions of dissimilarity of the 'Full Pretraining' group (based on Fig. 6b). p-values from T-test corrected for multiple comparison with fdr-bh.

| METRIC | UNTRAINED | MASTER & FROZEN | PARTIAL PRETRAINING | FULL PRETRAINING & UNFROZEN |
|---|---|---|---|---|
| DSA | $3.7 \times 10^{-4}$ | $2.0 \times 10^{-4}$ | $2.2 \times 10^{-1}$ | $6.3 \times 10^{-1}$ |
| CKA | $2.4 \times 10^{-1}$ | $1.5 \times 10^{-1}$ | $9.9 \times 10^{-1}$ | $8.2 \times 10^{-1}$ |
| Procrustes | $1.6 \times 10^{-1}$ | $5.5 \times 10^{-1}$ | $9.7 \times 10^{-1}$ | $6.7 \times 10^{-1}$ |

Table 10: Mamba: Comparing the distributions of dissimilarity of each training group against the distributions of dissimilarity of the 'Full Pretraining' group (based on Fig. 6a). p-values from T-test corrected for multiple comparison with fdr-bh.

| METRIC | UNTRAINED | MASTER & FROZEN | PARTIAL PRETRAINING | FULL PRETRAINING & UNFROZEN |
|---|---|---|---|---|
| DSA | $7.5 \times 10^{-1}$ | $2.7 \times 10^{-16}$ | $6.2 \times 10^{-4}$ | $4.0 \times 10^{-1}$ |
| CKA | $1.3 \times 10^{-26}$ | $4.8 \times 10^{-31}$ | $6.4 \times 10^{-7}$ | $2.2 \times 10^{-1}$ |
| Procrustes | $2.1 \times 10^{-26}$ | $4.7 \times 10^{-34}$ | $6.7 \times 10^{-6}$ | $4.5 \times 10^{-1}$ |

Table 11: Mamba: Comparing the distributions of dissimilarity of each training group against the distributions of dissimilarity of the 'Untrained' group (based on Fig. 6a). p-values from T-test corrected for multiple comparison with fdr-bh.

| METRIC | MASTER & FROZEN | PARTIAL PRETRAIN. | FULL PRETRAIN. | FULL PRETRAIN. & UNFROZEN |
|---|---|---|---|---|
| DSA | $1.2 \times 10^{-17}$ | $2.1 \times 10^{-3}$ | $7.5 \times 10^{-1}$ | $5.3 \times 10^{-1}$ |
| CKA | $4.6 \times 10^{-2}$ | $3.6 \times 10^{-15}$ | $1.3 \times 10^{-26}$ | $6.2 \times 10^{-29}$ |
| Procrustes | $1.5 \times 10^{-4}$ | $1.9 \times 10^{-16}$ | $2.1 \times 10^{-26}$ | $7.0 \times 10^{-27}$ |

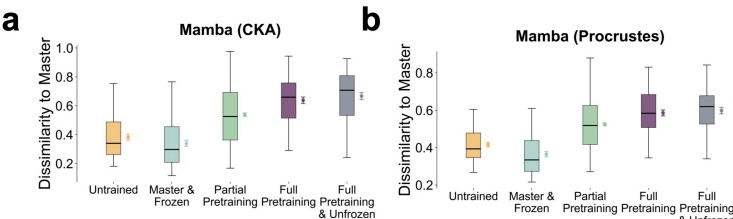

Figure 11: **CKA and Procrustes do not identify the reservoir-like learning in Mamba.** (a) Results from running the analysis from Fig. 3g with the Mamba architecture and CKA. (b) Results from running the analysis from Fig. 3g with the Mamba architecture and Procrustes.