# OpenReview forum: "Dynamical Similarity Analysis uniquely captures how computations develop in RNNs"
_ICLR.cc/2025/Conference — Submitted to ICLR 2025_

### Official Review · Reviewer_Kh9t · 2024-10-31

**Soundness:** 1
**Presentation:** 2
**Contribution:** 2
**Rating:** 3
**Confidence:** 5

**Summary:**

The authors investigate how three metrics, Dynamical Similarity Analysis (DSA), Procrustes distance, and linear Centered Kernel Alignment (CKA), quantify diffferences in RNN dynamics. They attempt to provide benchmarks, which illustrate favorable properties of DSA over the other two.

**Strengths:**

I think the intent and framing of the paper in the introduction is well written. The topic is interesting. Unfortunately, I have serious concerns about the execution of the project as outlined below.

**Weaknesses:**

This paper does not present a new technique or method, but is rather aiming to deepen our understanding of existing metrics. I really like this idea, but in my mind it means that the paper needs to be impeccably written and ideally involves some concrete mathematical results or guarantees about the methods being compared. I think the paper does not succeed on these terms and therefore does not meet the bar for publication at ICLR at its current level of polish and detail.

Let me summarize major weaknesses briefly:

* There are many places where the authors claim that there is an "expected" result that aligns with what DSA shows, but why these things are "expected" is not clearly described. I suspect that what one "expects" in many of these cases is debatable.
* The test cases shown are bespoke. It is not clear whether any of this generalizes to a broader variety of settings. There is also a relatively small number of tasks considered. I think they consider roughly 2-3 tasks, most being variants of Driscoll et al's study. In comparison, [Klabunde et al.'s recent benchmark](https://arxiv.org/abs/2408.00531) considers six different tests across six different datasets.
* There is a relatively small number of metrics considered. The authors consider three (DSA, CKA, and Procrustes). In comparison,  Klabunde et al's study linked above contains 23 similarity measures.
* Related to the point above, Procrustes and CKA were never meant to be applied to dynamical time series so the comparison seems a little unfair and expected that DSA comes out "ahead" in certain respects. At the same time, the authors do not include Diffeomorphic vector field alignment as a comparison to DSA (even though they do cite it). Additionally, I would point the authors to stochastic shape distances as a viable metric for comparing dynamical flow fields: [Lipshutz et al. (2024)](https://openreview.net/forum?id=Fykvxdv2I8)
    * For these reasons, the claim that "DSA **uniquely** captures" anything seems unjustified! I would only say that a method *uniquely* captures something if I had a mathematical proof that no other approach could work.


Below I unpack some of these weaknesses further with a bit more specificity:

* Regarding Fig 2C, there is no clear motivation why we would want a metric to respond "ratio-like" when we combine attractors. Furthermore, DSA only has marginally better linear R^2 (0.99 vs 0.97 or 0.96), yet this is somehow treated as a "win" for DSA over these other measures.
* Regarding Figure 3, the authors use the term "normative predictions" multiple times in relation to Driscoll et al.'s modeling work. I strongly encourage the authors to rephrase this. A normative model has a very specific meaning in theoretical neuroscience -- it involves predicting an attribute of a network on the basis of some functional or evolutionary principle (efficient coding is a classic example, see for e.g. [Mlynarski & Hermundstad, 2021](https://www.nature.com/articles/s41593-021-00846-0)). Driscoll et al. never use the term "normative model" in their paper and it is confusing to see the term applied here.
* Moreover, I am hestitant to treat the results of Driscoll et al. &mdash; which, while interesting, is only one empirical study of a very specific family of RNN tasks &mdash; as a foundational way to benchmark metrics on neural representations. The authors state at the conclusion of this section that "DSA is the only metric with correctly identifies the compositional representation that we expect." But it is not well explained what I should "expect" to see, and I suspect that what one "expects" to see could be debatable. In any case, the panel corresponding to DSA in Fig 3G does not seem to do a good job distinguishing the final 3-4 categories (only the yellow box plot seems substantially higher than the rest).
* The horizontal axis in Figure 2 is confusing. Epochs often refer to training epochs. The horizontal axis should be labeled "noise" or something similar.
* How noise impacts the simulation in Figure 2 is unclear. The authors don't show, for example, trajectories of neural firing rates. Also why plot only three levels of noise rather than a more fine scale grid?

**Questions:**

None

---

> ### Author Response · Authors · 2024-11-27
>
> Dear Reviewer,
>
> Thank you so much for taking the time to write detailed and constructive feedback on our work. In the following we outline the changes we implemented in response to your review. We hope that these changes, alongside the ones implemented for other reviewers, will alleviate at least some of the concerns you had about our manuscript.
>
> **From Weaknesses section**
>
> (1)
>
> *“There are many places where the authors claim that there is an "expected" result that aligns with what DSA shows, but why these things are "expected" is not clearly described. I suspect that what one "expects" in many of these cases is debatable.”*
>
> Across the manuscript we reworked sections to better explain how we derive our expectations. We hope that with our new explanations of expectations surrounding noise robustness (line 191-195), linear decrease in similarity over a linearly decreasing overlap in combined attractors (line 202-206), the similarity of RNN representations (line 310-320), and the link between representations and task solving abilities (line 361-368), our motivation for expectations will be more clear and uncontroversial. Generally speaking, our expectations for the attractor-based analyses come from carefully constructing test cases with simulated dynamics. The expectations for the RNN-based cases are derived based on prior publications on RNNs solving the Neurogym tasks. We hope our new sections explain more clearly how we arrive at expectations. We would be open to update our manuscript with a more nuanced perspective on what we should expect, if you outlined specific expectations which are contrary to the ones we use.
>
> In contrast to these clear predictions, we see sections 4.4 and 4.5 as exploratory sections where we apply DSA to new analyses to see if we can learn something about models from the test cases we constructed that goes beyond the a priori expectations. We agree that one could make different predictions about what these analyses ought to show and we hope that our comments in lines 400-403 and line 513-515 make clear that we do not see these analyses as means to test DSA or the other metrics.

---

> > ### Author Response · Authors · 2024-11-27
> >
> > (2)
> >
> > *“The test cases shown are bespoke. It is not clear whether any of this generalizes to a broader variety of settings. There is also a relatively small number of tasks considered. I think they consider roughly 2-3 tasks, most being variants of Driscoll et al's study. In comparison, Klabunde et al.'s recent benchmark considers six different tests across six different datasets.”*
> >
> > We of course cannot be completely sure that the shown abilities of DSA will generalise to new scenarios until we add a larger number of test cases. We now discuss this in lines 491-499. At the same time, we would find it very surprising if the shortcomings we identified in CKA and Procrustes do completely disappear in other scenarios, given that ultimately our tests were based on very standard RNN dynamics which will be observed across many other scenarios in the same way. As such, we feel confident that our results are prescriptive as far as to highlight that we should trust results derived with DSA more than results derived with CKA / Procrustes.
> >
> > We agree that our analyses are not as extensive as the work by Klabunde et al. in terms of numbers of tasks and metrics. We do want to note that while Klabunde and colleagues focus on benchmarking static representations, we focus on dynamic ones. The static representations have been explored for longer and hence we have gradually accumulated knowledge on how they could be benchmarked. We think that the case of static representations has shown how important it is to properly test these metrics but that for the dynamic case we have way less knowledge about what test cases could be used to construct a benchmark. We see our work as a first step into showing what test cases for dynamical representations that link to computations run over time series could look like. While we ‘only’ construct two completely independent test cases, one based on simulated attractors and one based on RNNs, we want to highlight that we run many independent simulations within each group. The attractor based simulations are based on 8400 independent comparisons and the RNN-based comparisons use 72 independent comparisons between each of the 11 group. We hope that our detailed analyses of both attractor- and RNN-based test cases will be a valuable basis for future developments in this area.
> >
> > With our test cases we do already show that DSA likely performs substantially better than static metrics adapted for dynamic use cases. Given that static metrics have been adapted to be used for dynamic use cases already (e.g. see cited work in lines 106-108), we think it is important to highlight that the better performance of DSA is likely worth the additional computational cost of computing the DSA metric.
> >
> > We hope that the reviewer agrees that our findings on DSA’s performance are worth sharing with the community, given that there clearly is an interest in adapting these metrics for dynamical use cases within the community.
> >
> > (3)
> >
> > *“There is a relatively small number of metrics considered. The authors consider three (DSA, CKA, and Procrustes). In comparison, Klabunde et al's study linked above contains 23 similarity measures.”*
> >
> > This is related to the comment above in (2), as Klabunde and colleagues could make use of a more extensive repertoire of metrics for static representations. We instead focused on DSA as a recently proposed metric with a public repository with a good working implementation, alongside CKA and Procrustes as they performed well in recent work by Cloos and colleagues (2024). The reason for not going beyond this relatively small set was that it already revealed that DSA shows all the trends we would have hoped a dynamic metric would show – namely the noise robustness, responses to combined attractors, responses to training schedule, and response to differences in accuracy. So we see the outcome of our investigations more in showing that DSA performs more reliably than these other metrics for these meaningful test cases and should hence be prioritised. We would be open to including additional static metrics in a future iteration of the manuscript if the reviewer had a specific metric that they see as more promising than Procrustes / CKA for the dynamic case.

---

> ### Author Response · Authors · 2024-11-27
>
> (5)
>
> *“For these reasons, the claim that "DSA uniquely captures" anything seems unjustified! I would only say that a method uniquely captures something if I had a mathematical proof that no other approach could work.”*
>
> We agree that the wording of the title was misleading. In response to this feedback and the feedback of a prior reviewer, we adapted our title and hope that it now reflects our contribution more accurately.
>
> **From 'Specific Weaknesses' section**
>
> (6)
>
> *“Regarding Fig 2C, there is no clear motivation why we would want a metric to respond ‘ratio-like’ when we combine attractors. Furthermore, DSA only has marginally better linear R^2 (0.99 vs 0.97 or 0.96), yet this is somehow treated as a ‘win’ for DSA over these other measures.”*
>
> Thank you for pointing this out to us as we believe we provided a poorly worded section in our prior manuscript. We now clarify that our main expectations for this section were (a) the noise robustness (lines 179-180 and 192-194) and (b) the linear decrease in dissimilarity to a linear decreasing influence of the additional attractor (lines 203-207 and 208-210). When interpreting the results we now specifically highlight that the differences between the metrics for this simple attractor-based test case are very minor and not meaningful per se (lines 215-218) – which is why we are then including the more complex RNN-based test case. We also highlight that the ‘ratio-like’ behaviour is more of an optional criteria but that we do not actually use it here to compare metrics (lines 208-210).
>
> We hope this will clarify that we do not see the attractor-based results as a ‘clear win’ for DSA. We do think they might point to a slight advantage of DSA, which might in turn crystallize in qualitatively better behaviour in the RNN-based test case on the side of DSA. If the only evidence we had were the attractor-based results, we would not advocate for DSA just based on these analyses.
>
> (7)
>
> *“Regarding Figure 3, the authors use the term ‘normative predictions’ multiple times in relation to Driscoll et al.'s modeling work. I strongly encourage the authors to rephrase this. A normative model has a very specific meaning in theoretical neuroscience -- it involves predicting an attribute of a network on the basis of some functional or evolutionary principle (efficient coding is a classic example, see for e.g. Mlynarski & Hermundstad, 2021). Driscoll et al. never use the term ‘normative model’ in their paper and it is confusing to see the term applied here.”*
>
> We agree this was a poor choice in wording. We have now removed the term ‘normative’ in this context and instead just refer to them as ‘predictions’ and ‘expectations’.

---

> > ### Author Response · Authors · 2024-11-27
> >
> > (8)
> >
> > *“Moreover, I am hesitant to treat the results of Driscoll et al. — which, while interesting, is only one empirical study of a very specific family of RNN tasks — as a foundational way to benchmark metrics on neural representations. The authors state at the conclusion of this section that "DSA is the only metric with correctly identifies the compositional representation that we expect." But it is not well explained what I should "expect" to see, and I suspect that what one "expects" to see could be debatable. In any case, the panel corresponding to DSA in Fig 3G does not seem to do a good job distinguishing the final 3-4 categories (only the yellow box plot seems substantially higher than the rest).”*
> >
> > As briefly discussed above, we did now expand our explanations of expectations with regards to the RNN-based analysis (lines 310-320). In this context we also adapted the visualisation of our prediction in Figure 3g which should now make clear why the pattern observed in DSA matches these predictions. In brief, they predict that pretraining on relevant subtasks induces representations which are similar to the ones developing when trained on the ‘Master task’. So when thinking about the similarity of different groups in comparison to networks trained on the ‘Master task’, we expect untrained networks to be the least similar, followed by partial pretraining, followed by full pretraining. These predictions are based on prior results and only manifest in DSA, not CKA or Procrustes.
> >
> > While your point is correct that the RNN-based test case strongly leans on results by Driscoll et al., we do want to point out that the tasks studied by Driscoll et al. have been studied by numerous authors before, so they are not the only ones that showed the effect of overlapping representations. For example Yang et al. 2019, NatNeuro (https://www.nature.com/articles/s41593-018-0310-2) showed related findings on composite representation using a different analysis method. We replicate the specific setup used by Driscoll et al. as we found that their detailed fixed point analysis gave the most detailed insight into the RNNs’ representations, but the predictions of compositionally overlapping representations are also backed by other work. We now added additional reference in lines 233-237 to highlight that there is a larger volume of work behind these tasks than just the work by Driscoll et al.
> >
> > We of course agree that this is not the only way by which we should benchmark dynamic metrics, but while the field has been relatively quick to use alignment metrics for dynamic use cases, we have not been good at coming up with ways on how to assess them. We believe that the compositional representations that develop in the context of Neurogym tasks provide us with a great opportunity, as prior work gives us a solid understanding of which representations should develop. We hope that this makes it more understandable why we believe the RNN-setup we chose can make for a valuable comparison of metrics.
> >
> > (9)
> >
> > *“The horizontal axis in Figure 2 is confusing. Epochs often refer to training epochs. The horizontal axis should be labeled "noise" or something similar.”*
> >
> > Thank you for pointing this out. We now use the word ‘Combinations’ instead, as we are not just varying noise levels in this analysis.
> >
> > (10)
> >
> > *“How noise impacts the simulation in Figure 2 is unclear. The authors don't show, for example, trajectories of neural firing rates. Also why plot only three levels of noise rather than a more fine scale grid?”*
> >
> > Thank you for highlighting that this could be clarified. The outcome of adding Gaussian noise to otherwise standard attractor dynamics that forms the basis of the attractor-based test cases is now visualised in Figure 8 in the appendix. We now also clarify in the text that the operation in this case is based on combining simulated attractors with sampled Gaussian noise, followed by scaling (line 184-187). We opted to plot three levels of noise as this allowed us to explicitly name / list the combinations in Figure 2a, and we found that this made the analysis easier to understand than using additional levels of comparison. At the same time, given that these analyses were fully controlled, we expected that intermediate intervals would follow a very predictable pattern in-between already sampled observations.
> >
> > We hope that the updates we made resulted in a more readable manuscript and can alleviate the concerns you had with regards to our initial submission. We want to thank you again for taking the time to give constructive feedback on our work!

---

> > > ### Comment · Reviewer_Kh9t · 2024-11-28
> > > **Reviewer Response**
> > >
> > > I thank and commend the authors for their hard work. After reading the other reviews and updated pdf, I feel most comfortable at keeping my score of 3. I think this work could be the starting point for a useful and impactful paper, but I ultimately do not think it is ready in its current form.
> > >
> > > I'm sorry for being a drag, but will try to leave some constructive comments in response to author comments below.
> > >
> > > > "...we think it is important to highlight that the better performance of DSA is likely worth the additional computational cost of computing the DSA metric. We hope that the reviewer agrees that our findings on DSA’s performance are worth sharing with the community..."
> > >
> > > I find this highly unpersuasive. The original paper that proposed DSA already made extensive comparisons to Procrustes distance and CKA scores. That paper showed, unsurprisingly, that these non-dynamical measures do not capture certain aspects of dynamical computation. The main contribution of the DSA paper was inventing a method that suceeeded where these methods failed.
> > >
> > > Any paper that follows up on DSA ought to substantially expand our understanding of that method (e.g. by deriving new mathematical theory) or change/extend the method itself. Every experiment in this paper strikes me as highly anecdotal.
> > >
> > > > DSA shows all the trends we would have hoped a dynamic metric would show – namely the noise robustness, responses to combined attractors, responses to training schedule, and response to differences in accuracy.
> > >
> > > As a first remark, I still think that the writing could benefit from more revisions to clarify why some of these behaviors of DSA are good and things that should be "hoped for" (e.g. training schedule is still confusing for me).
> > >
> > > Let's take noise robustness as a concrete example. First, this contradicts my direct experience with using DSA on simulated and real data. It is extremely sensitive to noise in my hands. Second, there is no noise model in DSA, so at best the authors are pointing out an anecdotal empirical result of DSA's useful robustness to noise. The authors don't provide any mathematical/theoretical explanation for this result, let alone a theorem that guarantees that DSA will "work" under certain noise conditions. I would view something along these lines as a minimum standard for reaching publication threshold.

---

### Official Review · Reviewer_sEeQ · 2024-11-01

**Soundness:** 3
**Presentation:** 3
**Contribution:** 3
**Rating:** 8
**Confidence:** 4

**Summary:**

This paper demonstrates that Dynamical Similarity Analysis (DSA), a recently introduced metric for comparing dynamical systems, outperforms related techniques such as Procrustes and CKA. DSA operates by projecting system trajectories into a high-dimensional space, where the vector fields governing the dynamics become approximately linear. It then aligns the two vector fields through an orthogonal transformation. The authors show that DSA consistently outperforms CKA and Procrustes across all tested tasks and suggest that DSA may be uniquely capable of capturing the computational processes underlying the tasks.

**Strengths:**

This paper is timely and addresses an important challenge in neuroscience and AI: comparing dynamic trajectories across neural systems. The paper makes a nice contribution by empirically evaluating several widely used techniques for comparing dynamical trajectories and identifying the most effective one as DSA. The comparisons appear to me to be done fairly and objectively.

**Weaknesses:**

The paper is at times quite hard to read, in particular when the authors are describing the tasks. The paper would benefit from a streamlining of the explanations of the experiments. Also (as I'll argue below), the title is a bit misleading, and should be changed to reflect the actual contributions of the paper.

**Questions:**

- The title suggests that you have shown/proved that DSA is unique among all similarity metrics in its ability to capture how computations evolve. You have not shown this. In reality, you have shown that DSA is better at two other metrics on a range of tasks. This is still an important contribution, but the title should be toned down to reflect the actual contributions of the paper.
- Is the appendix missing? There is a reference to it (L152), but it doesn't appear to go anywhere.
- L118: "... the momentum of traces in instead of..." I'm not sure what this is trying to say.

---

> ### Author Response · Authors · 2024-11-27
>
> Dear Reviewer,
>
> Thank you so much for your recommendations for improving our manuscript. These are the changes we implemented in response to your feedback:
>
> **From Weaknesses section**
>
> (1)
>
> *“The paper is at times quite hard to read, in particular when the authors are describing the tasks. The paper would benefit from a streamlining of the explanations of the experiments.”*
>
> We have made a large set of changes to better explain how we test for specific predictions across experiments and hope that this will make it easier to read. We are happy to make specific improvements if there was a particular section you thought could be improved.
>
> **From Questions section**
>
> (2)
>
> *“[...] but the title should be toned down to reflect the actual contributions of the paper.”*
>
> Thank you for that suggestion – you are right that the initial title did not strike the right tone and we changed it to specifically highlight that DSA can capture the specific kind of compositional dynamics that we are studying in our context. We think that this updated title is a more accurate reflection of the work. Please note that the updated title is "Dynamical similarity analysis can identify compositional dynamics developing in RNNs". The new title is given as a title in the manuscript but we believe this cannot be changed on OpenReview directly at the current stage.
>
> (3)
>
> *“Is the appendix missing? There is a reference to it (L152), but it doesn't appear to go anywhere.”*
>
> You are right that we are using an in-text link to the references file but as ICLR requires us to upload the appendix as a separate file under ‘Supplementary Material’, the in-text reference does not directly open the corresponding section in the document. Please refer to the ‘Supplementary Material’ section on OpenReview to find the relevant section in the appendix.
>
> (4)
>
> *“L118: ‘... the momentum of traces in instead of...;’ I'm not sure what this is trying to say.”*
>
> Thank you for pointing this section out to us. We meant this to say that DSA does not only consider the shape of the trace (i.e. which points in state space / representational space are connected) but also considers the momentum (i.e. speed) of transitioning between different points in state space / representational space. We now added this additional clarification to the relevant section in lines 110-113.
>
> We hope that the updates we made resulted in a more readable manuscript. We want to thank you again for taking the time to give constructive feedback on our work!

---

### Official Review · Reviewer_uRKZ · 2024-11-02

**Soundness:** 3
**Presentation:** 3
**Contribution:** 3
**Rating:** 5
**Confidence:** 4

**Summary:**

The paper has two main contributions. First, the authors suggest a benchmark framework in which to evaluate dynamic similarity metrics. Second, they use this framework to compare three specific metrics: Procrustes, CKA and DSA. They conclude that DSA is the best one. Finally, applying DSA to a more recent architecture MAMBA, they report little change to the dynamical component through training.
Specifically, the benchmark has two parts. The first is sampling from noisy ODEs and comparing between different noisy versions. The expectation is that the metric will be robust to noise, but capture genuine differences in the non-noisy dynamics. Further, when combining data from two ODEs, the authors expect a specific form of linearity.
The second part is using trained RNNs on a compositional task. Here the authors ask whether pretraining or constrained training induces representational similarities that are expected by task structure. These expectations are what makes RNNs a benchmark: task structure should induce similarity structure.

**Strengths:**

The paper addresses an important problem – comparing dynamical systems. It is valuable both for neuroscience and potentially for ML (e.g. MAMBA and other state space models). With the introduction of new metrics, there is a need for benchmarks to evaluate such metrics.
Using RNNs in a systematic manner is a good and original approach for achieving such a benchmark.
The authors simulate a large number of networks, while changing training schedule or task composition in a very systematic manner. This provides opportunities for teasing apart subtle differences between various metrics.

**Weaknesses:**

The results are somewhat preliminary. In particular, there are no insights or proofs on why there are differences between the metrics. The benchmark itself is rather qualitative.


1.	The paper lacks more rigorous expectations on how the benchmark results should look like. Why should we expect linearity in the attractor case? The RNN expectations are somewhat crude, as they only dictate whether one group is more dissimilar than another (also see point 2).
2.	All comparisons in the RNN are to the master network, and yet conclusions are drawn regarding their similarity to each other.
3.	Clarity of writing can be improved. There were several places where it was quite hard to understand what exactly was done.

**Questions:**

4.	If this benchmark is widely used, and metrics are optimized to be good at it. Will they be good metrics? In other words, how can we be sure that passing this benchmark generalizes to the broader objectives of metrics?
5.	The objectives of metrics are not fully elaborated. There is some mention of ratios, but this is not fully developed. This is not a trivial question, and not necessarily easy to answer. But it should be properly discussed in a paper suggesting benchmarks.
6.	The writing in the results section 4.1 is a bit jumpy. Definition of ratio. Then noise.
7.	Does the ‘+ and ‘1/2’’ notation in Fig 2a represent numerical addition and scaling, or a union of data sets from A/B/noise with different ratios? The main text (and appendices) failed to clarify this crucial point making subsequent results difficult to interpret with confidence.
8.	Results in Fig 3G are interesting. Why Procrustes and CKA can’t discriminate the untrained network? Is it possible that they do discriminate it, but because we are only measuring dissimilarity to the master, we can’t see it?
9.	Why do you compare groups and not individual networks?
10.	“The relative relevance of Attractor B” – number of trials? Amplitude?
11.	The Untrained network is a good control to have, as it illustrates how statistics of input can dominate dynamics and similarity measures.
12.	 Violin plots could be more informative than bars in the various plots.
13.	Figure 3: the expectations cartoon is confusing. The grey and purple have a specific order, despite the text saying they are expected to be the same. Furthermore, the order in the actual plots is opposite.
14.	The DSA paper uses dimensionality reduction and classification to quantify distances between networks. It could be useful to visualize all networks of figure 3 using multi dimensional scaling as in that paper. For instance, that might help understand whether pretrained networks are really similar to untrained networks (in CKA and Procrustes), or are simply equally distant from the master.
15.	Figure 4 – why are the values at 0 accuracy difference so distant from zero? Shouldn’t this include networks that are almost identical?
16.	Lines 429-431: are purple and green in the figure?
17.	Line 431: the pattern is inverted for zero percent. What about the decline from 70% training to 100% training?
18.	Line 467 “This means that all training groups produce roughly the same dynamics” – can this conclusion be reached when only comparing models to master, and not one to another?

---

> ### Author Response · Authors · 2024-11-27
>
> Dear Reviewer,
>
> Thank you so much for your recommendations for improving our manuscript. These are the changes we implemented in response to your feedback:
>
> **From Weaknesses section**
>
> (1)
>
> *“The paper lacks more rigorous expectations on how the benchmark results should look like. Why should we expect linearity in the attractor case? The RNN expectations are somewhat crude, as they only dictate whether one group is more dissimilar than another (also see point 2).”*
>
> We agree that the specific expectations / predictions of each benchmark were unclear in the prior version of the manuscript. For the attractor-based case we have now reworded the relevant section to specifically highlight why we expect a linear decrease in similarity (lines 203-206 and 209-210).
>
> For the RNN-based test case, we also now include two dedicated paragraphs explaining how our qualitative predictions derive from prior results (lines 265-309, lines 310-320). We agree that the qualitative predictions of this test case are less specific than the attractor-based ones. The increase in test complexity allowed for less specific predictions, however we do want to note that the RNN-based test case was successful in showing a difference in abilities across the metrics compared here. A full benchmark should have full quantitative predictions across multiple test cases but we believe that our investigations highlight a meaningful difference between metrics, and hence an important step towards testing dynamical representation metrics.
>
> (2)
>
> *“All comparisons in the RNN are to the master network, and yet conclusions are drawn regarding their similarity to each other.”*
>
> We now clarified in the prediction section of the RNN-based analysis (lines 320-325) that main analyses are run as comparisons to the ‘Master’ group. To address your point we now also added additional analyses where each group is compared to every other group (see appendix Figure 9). We discuss this figure in more detail in response to one of your later questions.
>
> (3)
>
> *“Clarity of writing can be improved. There were several places where it was quite hard to understand what exactly was done.”*
>
> We hope that stating our predictions more clearly and addressing yours and others’ points has helped make the manuscript more clear. We are of course grateful for any additional pointers on where the explanations are unclear, so that we can further improve the manuscript.
>
> **From Questions section**
>
> (4)
>
> *“How can we be sure that passing this benchmark generalizes to the broader objectives of metrics?”*
>
> We now added a section in the discussion (lines 499-504) addressing the question of generalization. Generally speaking, we do not think we can 100% be sure that the shown abilities of DSA will generalise to new scenarios until we add a larger number of test cases. At the same time, we would find it very surprising if the shortcomings we identified in CKA and Procrustes do completely disappear in other scenarios, given that ultimately our tests were based on very standard RNN dynamics which will be observed across many other scenarios in the same way. As such, we feel confident that our results are prescriptive as far as to highlight that we should trust results derived with DSA more than results derived with CKA / Procrustes.
>
> (5)
>
> *“The objectives of metrics are not fully elaborated. There is some mention of ratios, but this is not fully developed. This is not a trivial question, and not necessarily easy to answer. But it should be properly discussed in a paper suggesting benchmarks.”*
>
> We now make sure to make very specific and reasoned predictions with both the attractor-based and RNN-based test cases. As a specific note to your point about mentioning ‘ratios’: We agree that this was not properly explained and contextualised in the prior manuscript. We now elaborate how the linear decrease derives from the setup of the second attractor analysis (lines 203-207) and also how ratio-like behaviour could be a potential goal for metrics (lines 208-212), but that this is not actually a key assumption for our analyses.
>
> (6)
>
> *“The writing in the results section 4.1 is a bit jumpy. Definition of ratio. Then noise.”*
>
> We agree that the writing was unclear as we did choose odd locations to explain key terms. We now updated the attractor-based sections to first focus on the noise analysis and only later introduce the terminology that is relevant to the ‘linear decrease’ focused analysis. We hope this makes the section clearer and easier to understand.
>
> (7)
>
> *“Does the ‘+ and ‘1/2’’ notation in Fig 2a represent numerical addition and scaling,”*
>
> This notation means that we numerically added the attractors or the noise dynamics and then scaled them back to be contained in the unit volume. We modified the text in lines 184-186 accordingly.

---

> > ### Author Response · Authors · 2024-11-27
> >
> > (8)
> >
> > *“Why Procrustes and CKA can’t discriminate the untrained network? Is it possible that they do discriminate it, but because we are only measuring dissimilarity to the master, we can’t see it?”*
> >
> > As briefly discussed above, we now added a new analysis to address this question, as we agree it is an interesting one. Figure 3g indeed only showed the dissimilarity of all groups against the 'Master' group, and hence highlighted that different groups had the same representational distance to the Master group, but did not speak to the distance in-between groups. The additional analysis where we compare all groups against each other (Figure 9; referenced in lines 434-436) shows that it is not only that these two groups have the same representational distance to the ‘Master’ group but that, even when compared with each other, they show the same representations. This again highlights quite drastic shortcomings of CKA / Procrustes.
> >
> > (9)
> >
> > *“Why do you compare groups and not individual networks?”*
> >
> > We do actually compare individual networks within each group. When we compare two groups each containing 72 networks, we get 72 dissimilarity values corresponding to the 72 one-to-one network comparisons. The distributions plotted as boxplots show these 72 dissimilarities. We clarified this in lines 321-325.
> >
> > (10)
> >
> > *“The relative relevance of Attractor B” – number of trials? Amplitude?”*
> >
> > By relevance we refer to the amplitude of the attractor, and not the number of trials. We now clarified this in lines 199-202.
> >
> > (11)
> >
> > *“The Untrained network is a good control to have, as it illustrates how statistics of input can dominate dynamics and similarity measures.”*
> >
> > We agree that Untrained is a good control group as it shows how metrics can capture spurious signals. Our analyses specifically highlight that Procrustes and CKA are not good at discriminating untrained networks from trained ones. As such, our analyses highlight major shortcomings of these two metrics.
> >
> > (12)
> >
> > *“Violin plots could be more informative than bars in the various plots.”*
> >
> > We agree that showing the distribution of data underlying the plotted means is very informative. The ‘dissimilarity gap’ plots of Figure 2 so far did not have any distributional data, so we updated the figure with violin plots accordingly. The ‘linearity’ in the same plot is just a single value derived from data with regression, hence we do not show a distribution in this instance. Note that all other figures already highlighted the distribution of the data by including both boxplots and standard errors. We find these ‘parametric’ ways of highlighting the data distribution easier to understand for group comparison, and so we decided to use these across the paper. Note that, where additional regression analyses are run on top of the data (namely Figure 4b and 5c), we plot the raw data points or violin plots of data distributions, respectively.
> >
> > (13)
> >
> > *“Figure 3: the expectations cartoon is confusing. The grey and purple have a specific order, despite the text saying they are expected to be the same. Furthermore, the order in the actual plots is opposite.”*
> >
> > We agree that the expectation was not visualised in an ideal fashion. Alongside the expanded explanation in the text regarding the predicted order in the RNN tests (lines 310-320) we also updated the visualisation of the expectation (Figure 3g).
> >
> > (14)
> >
> > *“It could be useful to visualize all networks of figure 3 using multi dimensional scaling as in that paper. For instance, that might help understand whether pretrained networks are really similar to untrained networks (in CKA and Procrustes), or are simply equally distant from the master.”*
> >
> > As discussed in the context of question 8, we did add a new figure in the appendix (Figure 9) which highlights the representational distances between all groups, using the respective similarity metrics. We did consider further summarising these pairwise differences using multi dimensional scaling but ultimately found that the pairwise differences are a clear way of highlighting the similarities / differences across groups. If the reviewers felt strongly about MDS visualisations, we would be open to include these in a future update of the manuscript.

---

> > > ### Author Response · Authors · 2024-11-27
> > >
> > > (15)
> > >
> > > *“Figure 4 – why are the values at 0 accuracy difference so distant from zero? Shouldn’t this include networks that are almost identical?”*
> > >
> > > Your observation is correct and we now specifically allude to this fact in lines 378-381. We believe that this variance of representations for networks with the same accuracy comes from the fact that this analysis includes networks from across all training schedules / groups. What this analysis shows is that when networks from across groups reach higher accuracy and hence perform increasingly similar operations, their representations also increase in similarity. As they have learned the task in different ways, there still is some variance in the exact dynamic they use. It would be interesting to use future investigations to see which factors can exactly predict this variance. Please also note that our more controlled analyses based on attractors also highlights the issue surrounding DSA not reaching full zeros even with identical / near identical dynamics (discussed in lines 209-214).
> > >
> > > (16)
> > >
> > > *“Are the purple and green groups in the figure line 429-431?”*
> > >
> > > Yes they are (partial and full pretraining). We want to investigate the development of computations during the training phase, by grouping together the groups which have a similar ‘pretraining rank’, or a same number of tasks they were pretrained on. As such, the green group represents either 33% or 66% of rank (two tasks during pretraining at most) against 100% for the purple group, fully pretrained. We clarified this point lines 425-431.
> > >
> > > (17)
> > >
> > > *“What about the decline from 70% to 100% Line 431?”*
> > >
> > > The decline in dissimilarity (to the master group) is due to a better alignment for the fully pretrained group (100% rank) compared to the partially pretrained group (66%). The former, trained on all the subtasks that compose the master task, end up with a closer representation to master than the partially pretrained groups. We now highlighted this in lines 430-433.
> > >
> > > (18)
> > >
> > > *“ ‘This means that all training groups produce roughly the same dynamics’ – can this conclusion be reached when only comparing models to master, and not one to another?”*
> > >
> > > Thank you for pointing out that this wording was not completely accurate given our initial results. We believe that with the new analysis (Figure 9) that we discussed in the context of question 8 and 14, this statement is now backed up by our analyses. In Figure 9 (Appendix 6.3) we ran pairwise comparisons across all groups. What we see is that, for CKA and Procrustes, the groups are roughly at equal distance from each other, even when comparing very different training schedules. For example, Master and Master & Frozen are at a similar distance to Full Pretraining for both CKA and Procrustes.
> > >
> > > We hope that the updates we made resulted in a more readable manuscript and can alleviate the concerns you had with regards to our initial submission. We want to thank you again for taking the time to give constructive feedback on our work!

---

> > > > ### Comment · Reviewer_uRKZ · 2024-12-01
> > > >
> > > > I thank the authors for the extensive work in addressing reviewer comments.
> > > >
> > > > Figure 9 in the appendix is very helpful. I am somewhat surprised that CKA and Procrustes do not show any difference between untrained networks and trained networks. It is true that input is a strong driver, but still – trained networks often have a low-rank structure, and dynamics are restricted to lower dimensionality. I couldn’t find many details on how CKA and Procrustes were implemented. Whether dynamics had noise / multiple initial conditions. Are these averaged and then compared.
> > > >
> > > > Thanks for clarifying the + and 1/2 and numerical addition and scaling.
> > > > Adding noise 'makes sense' to simulate brain data: a noisy view of the same dynamics should be identified as the same dynamics. Is the addition of trajectories from two dynamical systems something you expect to see in real data?
> > > > Perhaps a more natural control to simulate a change in the underlying dynamics is just changes to the parameters resulting in bifurcations, etc.
> > > >
> > > > More generally, I was hoping for more intuition regarding the results. It is clear that DSA uses dynamics, and could therefore discriminate cases that a static measure cannot. But – the distribution of states in a diverging vs. converging dynamical system is still very different. For instance – why is the slope in Figure 4 negative for the static measures, rather than being zero or positive?
> > > >
> > > > Overall, I think that providing a benchmark for metrics is an important goal. Using compositional RNNs is a good and original approach to do so. Having mostly qualitative expectations, and no proofs/intuition on why the performance of DSA or the other metrics behaves in the way it does weakens the paper. I am thus keeping my score of 5.
> > > >
> > > > Line 328: DSA->Procrustes

---

### Official Review · Reviewer_59Jx · 2024-11-08

**Soundness:** 4
**Presentation:** 2
**Contribution:** 3
**Rating:** 5
**Confidence:** 4

**Summary:**

The work studies a recently proposed "Dynamical Similarity Analysis" (DSA) and benchmarks it against prior methods such as Centered Kernel Alignment (CKA) and Procrustes for analyzing the dynamical representations within neural networks, particularly in Recurrent Neural Networks (RNNs). Through systematic test cases that simulate both noise and compositional learning, the study argues that DSA provides a more robust and behaviorally relevant measure of dynamical alignment compared to established metrics like . Furthermore, the paper explores DSA’s application to new architectures such as Mamba models, suggesting that their internal dynamics operate differently from RNNs.

**Strengths:**

- The designed benchmark is novel and much needed to bridge the fields of computational neuroscience and mechanistic interpretability. Even though both fields look at similar problems, their languages have been very distant from each other for some time. I believe this work is a necessary step towards bringing them closer.

- The study of compositional learning in this context, as far as I am aware, is quite novel as well.

**Weaknesses:**

- The biggest weakness is the missing methods section. I see there is a supplementary file (which can be at the end of the original submission as an appendix), but this file does not contain the necessary information to reproduce these experiments. As a rule of thumb, by reading the methods section, without looking at the specific code, one should be able to reproduce the work. The public code is to help facilitate the process of reproduction, but is not a substitute for the writing. For example, what were the learning rates? How long were networks trained etc.?

- Though [1] is cited, I would have loved to see the method of finding and identifying the fixed points for categorizing the similarity of computation between RNNs as a baseline. I understand that not all problems will be solved by fixed points, but it is needed to show that DSA CAN recover the computational structure as efficiently in the benchmarks of [1]. For example, you can consider the 3-bit flip flop task and/or the sine generation task, in which we know the solutions and therefore can test whether DSA would be as effective as the fixed-point finders.

**Questions:**

I believe I understood most of the work adequately. My score is based on the current submission and the weaknesses described above. If you can address them, it is likely that my score will increase to match the strengths I described above.

[1] Maheswaranathan, Niru, et al. "Universality and individuality in neural dynamics across large populations of recurrent networks." Advances in neural information processing systems 32 (2019).

---

> ### Comment · Reviewer_59Jx · 2024-11-27
>
> As the reviewer of this work, I want to leave a public comment stating that I was truly surprised by author's decision to not write a rebuttal. I believed, and still do, that this work could have been eventually accepted at ICLR. Looking at other reviewers' comments, a solid rebuttal would have probably brought this work over the publication threshold and frankly be more respectful of our time as reviewers. I hope the authors will have stronger faith in their work in the future.

---

> ### Author Response · Authors · 2024-11-27
>
> Dear Reviewer,
>
> Thank you so much for your recommendations for improving our manuscript. We also want to thank you for the encouraging words in your follow up comment. We understand that it can be frustrating when authors do not reply to constructive criticism. As some of the other reviewers suggested additional analyses it took us a bit longer to update the manuscript and hence we are posting our rebuttal relatively late. We hope that the updates we made resulted in a more readable manuscript that alleviates the concerns you had with regards to our initial submission.
>
> These are the changes we implemented in response to your feedback:
>
> (1)
>
> *“The biggest weakness is the missing methods section. [...] For example, what were the learning rates? How long were networks trained etc.?”*
>
> We extended the section 3 (Methods) in the Appendix (6.1). We included all the necessary information to train the models (hyperparameters, loss function, validation metric, data size, number of epochs, training time, etc.) and to perform the analysis, as well as the hardware and software used to make it easier to reproduce the experiments. This information can be found in lines 722-828.
>
> (2)
>
> *“Though [1] is cited, I would have loved to see the method of finding and identifying the fixed points for categorizing the similarity of computation between RNNs as a baseline.”*
>
> We agree that this is an interesting analysis and luckily Driscoll and colleagues already include this analysis in the work that we base our RNN-based test case on. To clarify that this detailed analysis was already done in prior work, we now specifically add this information in lines 260-264.
>
>
> We want to thank you again for taking the time to give feedback on our work!

---

> > ### Comment · Reviewer_59Jx · 2024-11-28
> >
> > I appreciate the authors' last minute rebuttal, however I have made my plans according to the original deadline. For the future, please do not write a rebuttal in the last day of the posted deadline. I agree with the other reviewer that this work needs more work before publication, especially in terms of presentation, and little to no time to work it out with authors. My final recommendation is reject, though I wish the authors all the best in future venues

---

> > > ### Author Response · Authors · 2024-11-28
> > >
> > > Dear Reviewer,
> > >
> > > we are sorry that our answer to your points did not address your concerns. Given that the discussion period still lasts until the 3rd of December we were under the impression that we could still engage in discussion now but we appreciate that you made your plans according to the deadline of posting last updates to the uploaded manuscript.
> > >
> > > Thank you again for taking the time to give feedback on the manuscript and we of course would be glad to take any further feedback into account.

---

### Author Response · Authors · 2024-11-28
**Comment for all reviewers**

Dear Reviewers,

We want to thank you all for taking the time to give constructive criticism on our manuscript. Here we want to give a brief comment on larger changes that we implemented which are relevant to all reviewers’ comments:

**(1)** We updated the title of the manuscript to now read “Dynamical similarity analysis can identify compositional dynamics developing in RNNs” in order to more specifically represent our contribution.

**(2)** We added new analyses to the RNN-based section which now not only considers similarity comparison with the ‘Master group’ as reference but instead we now provide values for all pairwise group comparisons in Figure 9. The interpretation of the results did not change as a function of these new analyses as the new results also highlight that CKA and Procrustes struggle to identify fundamental differences in network training, while DSA is able to identify them. We thank the reviewers for suggesting the addition of these analyses.

**(3)** Across both the attractor-based and RNN-based tests we acknowledge that the initial manuscript used somewhat convoluted wording with regards to the predictions made for each test case. We now provide additional details on every prediction we made that we expect metrics to adhere to and hope that this makes the evaluation of metrics easier to understand. This also includes an updated visualisation for predictions of the RNN-based analyses shown in Figure 3.

**(4)** As suggested, we significantly expanded the methods section with details on our implementation of network models and tasks, alongside specific examples and visualisations of the dynamics which formed the foundation of attractor-based analyses.

Overall, we think that our manuscript has been improved and we thank the reviewers for their time and efforts.

---

### Meta-Review · Area_Chair_eXkg · 2024-12-24

**Metareview:**

This submission compares similarity metrics in neural networks in dynamic settings, highlighting the advantages of Dynamical Similarity over other methods like Procrustes and CKA. While the topic is important, timely, and the intent of bridging neuroscience and interpretability is commendable, the reviewers have identified weaknesses that prevent its acceptance at this stage.

**Additional Comments On Reviewer Discussion:**

Reviewers noted that the test cases, while interesting, are bespoke and do not convincingly generalize to broader settings, and the paper lacks mathematical rigor or theoretical guarantees to support its claims. Furthermore, the comparisons against non-dynamic methods like CKA and Procrustes seem misaligned with the core strengths of DSA, raising concerns about the fairness and relevance of the benchmarks. The writing, particularly in describing tasks and motivating expectations, is often unclear, making it difficult to assess the robustness of the methodology. While the authors' extensive revisions and rebuttals addressed some concerns, the fundamental issues remain.

---

### Decision · Program_Chairs · 2025-01-22

Reject